# Microfinance, an Alternative for Financing Entrepreneurship: Implications and Trends-Bibliometric Analysis

Katherine Coronel-Pangol [1,*] 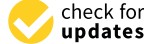, Doménica Heras-Tigre [2], Jonnathan Jiménez Yumbla [1], Juan Aguirre Quezada [1] and Pedro Mora [1]

1  Department of Economics, Business and Sustainable Development, University of Cuenca,
   Cuenca 010150, Ecuador; jonnathan.jimenezy@ucuenca.edu.ec (J.J.Y.); juan.aguirreq@ucuenca.edu.ec (J.A.Q.);
   pedro.mora@ucuenca.edu.ec (P.M.)
2  School of Economics and Administrative Sciences, University of Cuenca, Cuenca 010150, Ecuador;
   domenica.herast@ucuenca.edu.ec
*  Correspondence: katherine.coronelp95@ucuenca.edu.ec

**Abstract:** Microfinance has become one of the most important financing alternatives for business start-ups, especially for vulnerable groups in poor regions. Microfinance provides access to financial products, especially to people who have been excluded from the traditional financial system. However, the mainstream literature on microfinance shows its impact on poverty alleviation, but it is not yet well developed to understand its dynamizing role in the entrepreneurial sector. There is still a large gap in the literature on analyzing microfinance as a financing alternative, so this paper seeks to find this relationship in the literature. A bibliometric analysis is applied, both of the performance of the publications and a word co-occurrence analysis during the period 2017–2022. The articles indexed in the Web of Science have been considered and systematized through the SCIMAT software v1.1.04, developed by the Soft Computing and Intelligent Information Systems Research Group, University of Granada, Granada, Spain. Microfinance institutions, education, entrepreneurship, organizational performance, business microcredits, and women microentrepreneurs have been identified as driving themes to be considered in future analyses. At the end of the document, the proposed future lines of research are presented. In addition, the results show the growing interest of the academic community in the topic, with 2022 being the year with the highest number of articles published on the topic.

**Keywords:** microfinance; entrepreneurship; bibliometric analysis; co-occurrence; SCIMAT

## 1. Introduction

Entrepreneurial activity has been analyzed for many years, but one of the issues of constant concern is access to finance (Agarwal and Pokhriyal 2022). One of the main obstacles, especially for entrepreneurs in their early stages, is obtaining financial resources that allow them to start and develop their activities (Nair and Njolomole 2020). In this regard, the literature has focused on identifying and linking financing alternatives that allow businesses to grow and become sustainable (Ranabahu and Tanima 2022). One such alternative is microfinance. Coelho et al. (2022) point out that microfinance emerges as a tool that promotes access to finance for people excluded from traditional financial systems, thus encouraging the development of entrepreneurial activities.

Microfinance was first introduced in the 1970s when Dr. Muhammad Yunus of the Grameen Bank, who won the Nobel Prize in Economics in 2006, began to provide microcredit as an alternative to eradicating poverty in Bangladesh (Yunus 2006). Its development has been analyzed mainly from the point of view of this approach as a primary tool to support the alleviation of poverty in the world (Mustafa et al. 2018; Coelho et al. 2022).

Microfinance promotes entrepreneurship, which is fundamental to alleviating poverty (Santos and Neumeyer 2021). On the other hand, Klapper et al. (2016) highlight that microfinance has led to increased household welfare, while Mustafa et al. (2018) emphasize

that microfinance is one of the economic innovation tools to combat poverty. This objective is also mentioned by Coelho et al. (2022), who also highlighted the empowerment of the most vulnerable. Bros et al. (2022) point out that the impact of microfinance is greater in regions with higher levels of poverty.

However, there are other aspects from which microfinance is studied. For example, Mustafa et al. (2018) highlight that one of the main trends and challenges of microfinance is the technological advances, which have made it possible to expand the coverage of its services. Coelho et al. (2022) analyze the main trends in microfinance and focus on the dichotomy of microfinance institutions between their social objective and their financial institutionality, financial innovation, support for entrepreneurship, poverty alleviation, and the development of tools, such as crowdfunding. Singh et al. (2021) emphasize that microfinance has the ability to adapt loans to the financial needs of individuals, unlike traditional financing, so this financial alternative adequately accompanies entrepreneurs. Rohman et al. (2021) highlight the importance of microfinance as the main alternative solution for small businesses and enterprises through its microcredit tools, given that bank financing has not been able to provide a solution to these problems.

Other studies, such as that of Mustafa et al. (2018), show that in recent years, microfinance has no longer focused exclusively on granting microcredit but has updated and innovated its services, mainly through digital tools and access to information technologies. In this regard, Bros et al. (2022) emphasize that it is important to understand the impact of microfinance in general, focusing on institutions that support entrepreneurship rather than just looking at the impact of microcredit as a financing tool. In addition, they analyze the impact of access to microfinance institutions on the likelihood of becoming an entrepreneur and show that living near one of these institutions is strongly associated with an increase in the likelihood of becoming an entrepreneur. This relationship is particularly strong in regions with higher levels of poverty.

Coelho et al. (2022) conducted a scientometric analysis and a comprehensive literature review to identify trends in entrepreneurship outcomes, analyzing more than 500 papers indexed in the Web of Science. The results show, among other things, the impact of microfinance on the economy, social development, group lending, cooperation networks, social capital, poverty reduction, entrepreneurial activities, innovation in financial services, and gender. For their part, the results of Menne et al. (2022) highlight access to microcredit, poverty alleviation, sustainability, and innovation in financial technologies, as part of the main areas of analysis and trends for the future.

In this sense, it can be seen that microfinance analyses are carried out in a general way, demonstrating its contributions and benefits, but there is no detailed analysis exclusively on the relationship between microfinance and entrepreneurship. For this reason, the relationship between the literature on these two topics is analyzed through a bibliometric study in order to identify the trending topics that are developing, as well as to propose future lines of research. Considering that the main characteristic and objective of microfinance is the eradication of poverty, this paper aims to make an original contribution by analyzing the impact of microfinance exclusively on entrepreneurship and to identify new research topics in this area of analysis. To this end, we work with the scientific papers of the last 5 years (2017–2022) and use the SCIMAT software v1.1.04, developed by the Soft Computing and Intelligent Information Systems Research Group, University of Granada, Granada, Spain. Thirteen key topics have been identified, which will be analyzed in detail in the development of this work. The results are important for microfinance institutions, entrepreneurs who choose or seek this financing alternative, decision-makers, and public policy-makers, among others.

This paper is structured as follows: after the introduction, the materials and methods are detailed, then the results are presented, followed by the discussion, and finally, the conclusions are presented.

## 2. Materials and Methods

For the development of the present work, an exhaustive search was carried out in one of the most important worldwide recognized databases, the Web of Science (WoS). The search equation and other relevant aspects are detailed in Table 1. The open-source software Sci-MAT, v1.1.04, developed by the Soft Computing and Intelligent Information Systems Research Group, University of Granada, Granada, Spain, was used to manage the information and develop the bibliometric analysis. This software was chosen for its flexibility and simplicity, with a selection of measures that allow the extraction and visualization of bibliometric networks and scientific knowledge maps, as well as their longitudinal analysis.

**Table 1.** Search indicators.

| Database | Web of Science |
|---|---|
| Date of consultation | March 2023 |
| Search equation | "entrepreneu*" and "microfinanc*" |
| Search periods | 2017–2022 |
| Type of document | Articles |
| Type of magazine | All types of journals belonging to the indicated bases. |
| Total items | 486 |

Note: In the search equation, the "*" allows finding broad terms considering the root indicated; for example, in "entrepreneu*", the terms entrepreneur, entrepreneurs, entrepreneurship were also considered; in the case of "microfinanc*", terms, such as microfinance, microfinancing, microfinances, or similar were also included.

Thus, the methodological process of bibliometric analysis consists of three phases: 1. In the first phase, a descriptive analysis was carried out to evaluate the scientific production in the areas of knowledge studied, which were studied using the WoS databases as a reference. In this phase, bibliometric performance indicators were analyzed, i.e., the number of published papers, most cited papers and authors, and H-index, among others; 2. A second phase consisted of the normalization of keywords, followed by their grouping into themes through the algorithm of simple centers. Finally, a strategic diagram of the topics or scientific map was obtained, which made it possible to locate them according to their centrality (*x*-axis) and density (*y*-axis). Scientific maps are spatial representations of the relationships between documents according to the criteria analyzed (Cascón-Katchadourian et al. 2020). The SCI-MAT (Science Mapping Analysis Software Tool) software, developed by Cobo et al. (2011), a research group in Soft Computing and Intelligent Information Systems at the University of Granada, Spain, was used for the elaboration of the science maps. Only an analysis period corresponding to the last 6 years has been considered, due to the exponential growth of the relevant literature during this period. Table 1 summarizes the search indicators used in this analysis.

According to the methodology of Cobo et al. (2011), based on a co-occurrence analysis and the H-index (Hirsch 2005) for the development of bibliometric work, the following phases were considered for the analysis of a research field:

The detection of research topics through word co-occurrence analysis, summed with keyword clustering using a simple center algorithm that locates networks of keywords that are closely related to each other. This similarity between keywords is evaluated using the equivalence index

$$e_{ij} = \frac{c_{ij}^2}{c_i c_j} \qquad (1)$$

where $c_{ij}$ is the number of documents, in which two keywords i and j coexist, and $c_i$ and $c_j$ are the number of documents, in which each keyword appears individually (Coulter et al. 1998).

Visualization of research topics. In this phase, the main visualization tool used is the strategy diagram, which consists of centrality and density measures. Centrality measures the degree of interaction of a network with other networks, while density measures the internal intensity of the network (Cascón-Katchadourian et al. 2020).

Figure 1 shows the structure of the strategic diagram and its representation in each quadrant, as detailed below. In the upper right quadrant are the driving themes, i.e., those with high centrality and density. They correspond mainly to well-developed and important issues. In the upper left quadrant are the themes with high density but low centrality, i.e., they are specialized but isolated themes. These themes are important because they are internally developed, but they have a reduced relationship with the other themes. In the lower left quadrant, there are themes with low density and low centrality, known as emerging or disappearing themes. These are internally underdeveloped themes with a weak relationship to the other themes. Finally, in the lower right quadrant are the high-centrality, low-density themes, which are not very developed but have a strong relationship with the other themes. These are generally cross-cutting, general, and fundamental themes (Cobo et al. 2011).

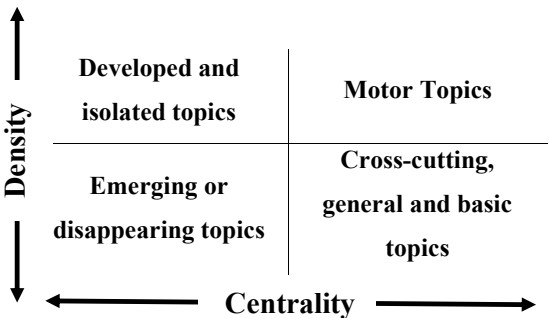

**Figure 1.** Strategic diagram.

### 3. Results

*3.1. Analysis of Scientific Production Performance*

The results have been analyzed from two approaches; on the one hand, as mentioned in the methodology, an analysis of the performance of publications has been carried out through descriptive tools of the main indicators.

By analyzing the number of articles published, it can be seen that in the last three years, there has been an increase in the production and analysis of microfinance and entrepreneurship. In the last year, more than 100 articles have been published on the analyzed topic.

In general, in the period 2017–2022, after a process of debugging and cleaning the initial database, 486 articles related to microfinance and entrepreneurship were identified, among which a linear upward trend line can be observed (Figure 2), with an average annual growth of 11%, proving the growing interest of the scientific community in this specific subject.

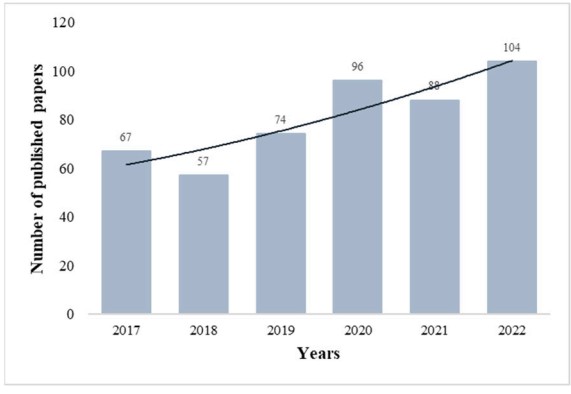

**Figure 2.** Scientific production by year.

Table 2 shows the authors with the highest scientific production in the subject analyzed, as well as the number of citations they have. It can be seen that Arvind Ashta from the Burgundy School of Business of Dijon is the author with the highest scientific production on this topic, with 14 published articles, followed by Jonathan Kimmitt from the Durham University Business School, who is the second author with the highest scientific production and the first with the highest number of citations. The H-index of each author was also taken into account.

**Table 2.** Author performance.

| Author | Publications | Citation | H-Index |
|---|---|---|---|
| Ashta, Arvind | 14 | 90 | 26 |
| Kimmitt, Jonathan | 5 | 99 | 14 |
| Al Mamun, Abdullah | 5 | 25 | 38 |
| Mahmood, Samia | 4 | 82 | 11 |

Similarly, the results of this analysis show that articles are published in 267 journals that deal with the subject. Table 3 shows the journals with the highest number of articles published, representing 15% of the total production analyzed. The main journal is *Strategic Change—Briefings in Entrepreneurial Finance*, which has more than 7% of the papers analyzed in this document.

**Table 3.** Journals with more publications in the area.

| Journal | Articles | % of Total |
|---|---|---|
| *Strategic Change—Briefings In Entrepreneurial Finance* | 35 | 7% |
| *Sustainability* | 11 | 2% |
| *Journal of Business Venturing* | 10 | 2% |
| *Small Business Economics* | 9 | 2% |
| *Journal of Business Ethics* | 9 | 2% |

Table 4 shows that the most cited article, with 152 citations, corresponds to Rawhouser et al. (2019), who examine conceptually and empirically the social impact of 71 social ventures on social entrepreneurship, determining as one of the main conclusions the need and importance of financing through collaborative tools, such as microfinance. On the other hand, Josefy et al. (2017), the authors of the second article with the second highest number of citations, analyze the role of the community in the success of the collaborative tool, such as crowdfunding, using as an example crowdfunding campaign to save a local theater, concluding that the active participation of the local community and the cultural identification with the theater were important factors in the success of the crowdfunding campaign. The authors highlight crowdfunding as one of the alternatives to microfinance. Certainly, that article provides useful information for entrepreneurs seeking to finance their initiatives through crowdfunding. Banerjee and Jackson (2017), on the other hand, analyzed the impact of microfinance as a means to reduce poverty in three communities in Bangladesh.

**Table 4.** Most cited articles in the field.

| Journal | Title | Authors | Year | Citations |
|---|---|---|---|---|
| Entrepreneurship Theory And Practice | Social Impact Measurement: Current Approaches and Future Directions for Social Entrepreneurship Research | Rawhouser, H., Cummings, M., Newbert, S.L. | 2019 | 152 |
| Entrepreneurship Theory And Practice | The Role of Community in Crowdfunding Success: Evidence on Cultural Attributes in Funding Campaigns to "Save the Local Theater" | Josefy, M., Dean, T.J., Albert, L.S., Fitza, M.A. | 2017 | 108 |
| Human Relations | Microfinance and the business of poverty reduction: Critical perspectives from rural Bangladesh | Banerjee, S.B., Jackson, L. | 2017 | 93 |
| Entrepreneurship Theory And Practice | Necessity or Opportunity? The Effects of State Fragility and Economic Development on Entrepreneurial Efforts | Amoros, J.E., Ciravegna, L., Mandakovic, V., Stenholm, P. | 2019 | 84 |
| Asia Pacific Journal Of Management | Economic growth, innovation, institutions, and the Great Enrichment | Ahlstrom, D., Tomizawa, A., Zhao, L., Bassellier, G. | 2020 | 65 |

*3.2. Analysis of Strategic Diagrams*

After the evaluation and analysis of the performance of scientific production in the field of microfinance and entrepreneurship, an analysis of related thematic studies was carried out by identifying the co-occurrence of keywords, considering a time horizon of 2017–2022.

For the period under analysis, terms with a minimum frequency of two and with a minimum frequency of co-occurrence of one were taken as reference, and the clusters obtained are the following: Microfinance Institutions, Entrepreneurship, Business Microcredit, Organizational Performance, Women Microentrepreneur, Education, Crowdfunding, Economy, Ecology, Youths, Capital Structure, Small and Medium Enterprises (SME's), Decision-Making, which can be seen in Table 5 with their respective ranges of centrality and density.

**Table 5.** Most cited articles in the field.

| Topics | Centrality | Density |
|---|---|---|
| Microfinance Institutions | 1 | 1 |
| Entrepreneurship | 0.77 | 0.85 |
| Business Microcredit | 0.85 | 0.69 |
| Organizational Performance | 0.69 | 0.92 |
| Women Microentrepreneur | 0.62 | 0.54 |
| Education | 0.92 | 0.77 |
| Crowdfunding | 0.38 | 0.62 |
| Economy | 0.46 | 0.46 |
| Ecology | 0.54 | 0.38 |
| Youths | 0.31 | 0.23 |
| Capital Structure | 0.23 | 0.31 |
| SME's | 0.15 | 0.15 |
| Decision-Making | 0.08 | 0.08 |

The composition of the strategic diagram (Figure 3) is interesting, as it can be seen that the themes found are mainly aligned between the upper right quadrant (driving themes)

and the lower left quadrant (emerging or disappearing themes). There is only one theme in the quadrant of developed and isolated themes, and only one in the quadrant of basic and transversal themes, but the position of both is very close to the union of the axes. The strategic diagram is presented as a function of the h-index, where the driving themes are those with the highest h-index, while those of the emerging and disappearing themes have the lowest h-index on average.

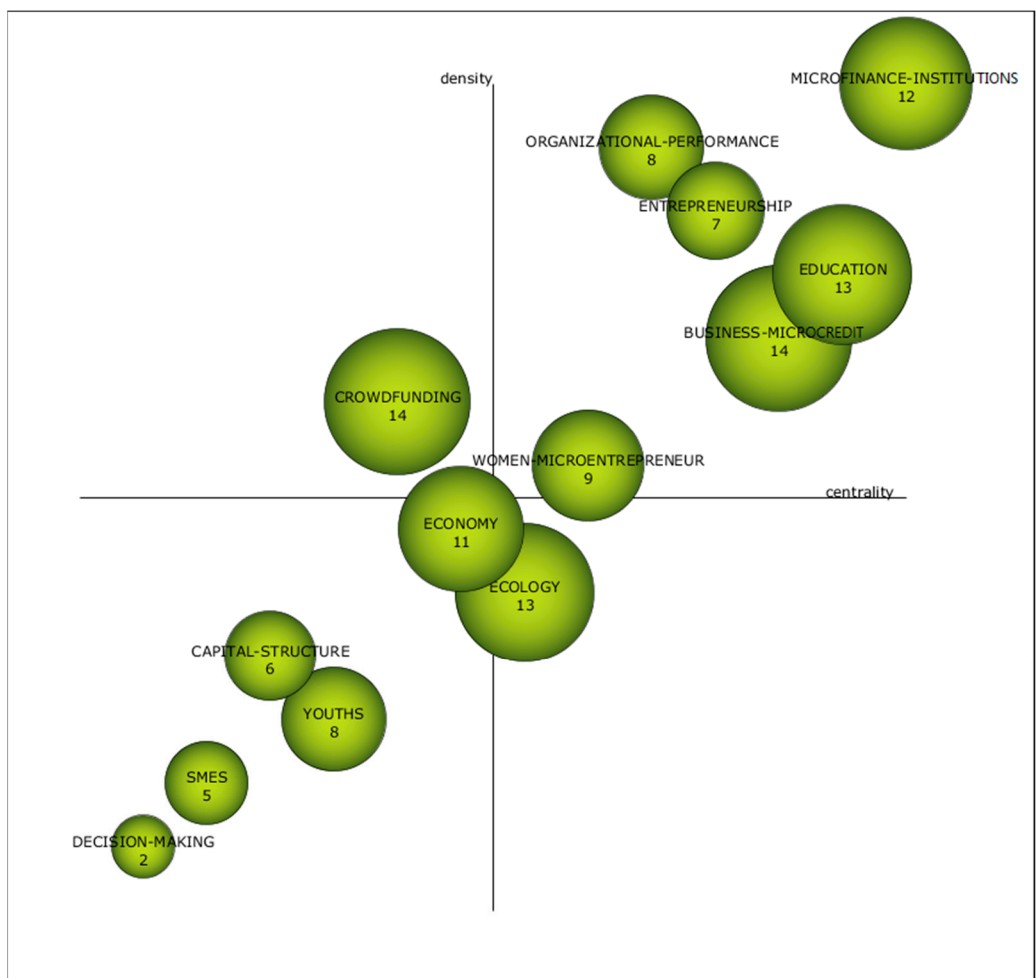

**Figure 3.** *Strategic diagram* 2017–2022.

The driving themes of the strategic diagram are observed and analyzed below:

- Microfinance Institutions. This cluster is located at the highest level of centrality and density, i.e., it is the most developed and related to a situation that is mainly explained by the theme analyzed. One of the main topics addressed by the authors when they talk about microfinance is financial institutions. Doshi (2010) believes that microfinance is an effective means to reduce poverty and that in order to achieve the desired social impact, it is essential that microfinance institutions are sustainable. Lam et al. (2019) argue that microfinance institutions are hybrid organizations with a dual mission of social purpose and financial sustainability. Shkodra et al. (2021) measure the role of microfinance institutions in the development of women's entrepreneurship and conclude that loans provided by microfinance institutions certainly have a positive impact on women's business and entrepreneurship performance. Bros et al. (2022) find that microfinance institutions promote financial inclusion and, thus, the economic development of countries, but they emphasize that the literature has focused only on microcredit and not on the impact that microfinance can have. They suggest that the nearby geographic presence of microfinance institutions increases the likelihood that

entrepreneurs will improve their businesses, especially in poorer regions. In this way, the importance of microfinance institutions can be appreciated.

- Education. This cluster is directly related to other clusters, such as financial inclusion and social impact, and plays an important role in the development of microfinance and entrepreneurship because, in several cases, people and entrepreneurs who seek financing through microfinance do not have adequate financial education for proper decision-making. In this regard, Perossa and Gigler (2015) points out that microfinance institutions plan to develop microsaving and microcredit systems to reduce the economic impact of inequality and lack of opportunities, although support for training and education is minimal.
- Business microcredit. Business microcredit is one of the issues directly related to banking. Gámez-Gutiérrez and Saiz-Álvarez (2011) mentions that microfinance is becoming a new financing system for entrepreneurs, especially in the third and fourth world, i.e., in developing areas, and, therefore, represents a valuable tool for entrepreneurs who do not have access to other types of financing. Yunus (2010), in his book "The Banker to the Poor", highlights that microcredit can be a safer and fairer alternative for people and entrepreneurs who do not have access to traditional financial services and are usually forced to resort to informal lenders.
- Women microentrepreneurs. This cluster has a moderately strong relationship with the human capital cluster. One of these is the study by Parvin et al. (2012), which examines the educational factors, access to resources, prior entrepreneurial experience, and family and community support that influence the development of women's microentrepreneurship in rural Bangladesh. Based on 410 interviews with women entrepreneurs, the results show that education and prior entrepreneurial experience have a positive impact on women's microenterprise development. Access to financial, physical, and social resources, as well as family and community support, were also found to be important factors in women's entrepreneurial success. Santos and Neumeyer (2021) also analyze women's microentrepreneurship, highlighting that it is one of the main alternatives for women to combat poverty, which mainly affects them. They also mention that microfinance promotes women's entrepreneurship.
- Organizational performance. This is a high-density issue that is considered to be a driving issue related to clusters, such as stakeholders and government support. Organizational performance is an important topic for business success as it enables, through strong organizational culture, the achievement of goals of employees in the organization, thus increasing the overall performance of the organization (Shahzad et al. 2012).
- Entrepreneurship. This topic is reasonably considered a driving theme and is associated with clusters, such as wealth distribution and legitimacy. Entrepreneurship empowers people by enabling them to pursue their dreams and generate new ideas that fill the gaps in the market. There are different types of entrepreneurships, such as social, technological, sports, and international, which provide an opportunity for ideas to become a reality (Ratten 2020).

The driving themes, which partly coincide with the literature reviewed, are microfinance institutions as logical spaces in which this activity takes place. In this sense, it is important to highlight new trends, such as hybrid institutions, combining the social approach with the need for financial sustainability. On the other hand, business microcredits are the main product of microfinance institutions; however, the literature suggests not to analyze them in isolation but in conjunction with all the other financial products that can be presented. One of the most interesting topics is that of women's microentrepreneurship, as the literature suggests a high relationship between women and microfinance since they are the main beneficiaries.

The basic and cross-cutting theme identified is Ecology. This cluster is related to social management and financing; it is a theme with a centrality rank of 0.54 that refers to ecological and environmental enterprises and businesses, which can be an important

element to support the country in its quest for a stable economy, a favorable social situation and environmental security (Valeryanovna 2012).

The high-density theme obtained is Crowdfunding. This theme is found to be associated with psychological factors and technology. Although the theme is in the peripheral theme quadrant, it is an important cluster as microfinance, crowdfunding, and lending have expanded and become visible among entrepreneurs in recent years as a tool and mechanism to access alternative financing to sustain and grow their businesses (Bruton et al. 2015). Crowdfunding is based on connecting entrepreneurs and funders through websites. Entrepreneurs and projects receive 41.3% of total crowdfunding, equivalent to $6.7 billion, while social causes receive 18.9% or $3.06 billion (Acconcia 2015).

Finally, the declining and/or emerging themes identified are as follows:

- Economy. This cluster is related to poverty and emerging economies, and according to the analysis of the chart, it is in decline. However, Barguellil and Bettayeb (2020) conducted a study on 114 clients of a microfinance institution to evaluate the impact of microfinance on economic development, from which they concluded that microfinance loans have a positive impact on economic development and the creation of new businesses because they generate employment and improve the economic situation of their clients. Therefore, microfinance is fundamental to the creation of entrepreneurship and economic growth in emerging economies;

- Youth. The youth cluster is associated with financial innovation and rural development. Talking about youths refers to the population group that plays a fundamental role in the development of entrepreneurship, which is essential to having solid entrepreneurial human capital (Kantis 2016);

- Capital structure. This topic refers to the capital structure and consists of the clusters of nonprofit organizations and business financing. In this regard, authors, such as Hăpău (2018), examined the capital structure of microfinance institutions and concluded that the capital structure had a significant impact on the financial performance of microfinance institutions. Specifically, that study found that higher levels of debt had a negative effect on financial performance, while higher levels of capitalization had a positive effect. Similarly, the study found that the age of the institutions and their size also influenced their financial performance. Older and larger institutions tend to have better financial performance;

- SMEs. This nomenclature refers to small and medium-sized enterprises and is associated with competitive advantage and microcredit. According to several authors, SMEs are fundamental to the growth of emerging economies and the relevance of microfinance credit lines to their financial performance (Woldie et al. 2012; Amsi et al. 2017);

- Decision-making. This topic refers to the decision-making of entrepreneurs in accessing microfinance for their entrepreneurial initiatives and is formally understood as the process that allows people to choose one alternative or another and applies to any circumstance, whether simple or complex (Robbins 1987). According to Sadler-Smith (2016), decision-making is a fundamental process for entrepreneurial success and is somehow associated with the intuition of entrepreneurs. This article highlights the importance of understanding and using intuition in the decision-making process of entrepreneurs.

After analyzing each of the issues in the strategic diagram, it is appropriate to evaluate both the productivity and the impact of each issue, which can be seen in the following table (Table 6).

**Table 6.** Impact and productivity by topic period 2017–2022.

| Topics | Number of Documents | H Index | Average Number of Citations |
|---|---|---|---|
| Microfinance Institutions | 44 | 12 | 11.84 |
| Entrepreneurship | 19 | 7 | 6.84 |
| Business Microcredit | 61 | 14 | 8.59 |
| Organizational Performance | 22 | 8 | 8.55 |
| Women Microentrepreneur | 41 | 9 | 9.07 |
| Education | 68 | 13 | 7.78 |
| Crowdfunding | 41 | 14 | 12.9 |
| Economy | 36 | 11 | 12.14 |
| Ecology | 25 | 13 | 20.84 |
| Youths | 12 | 8 | 15 |
| Capital Structure | 12 | 6 | 9.08 |
| SME's | 10 | 5 | 5.5 |
| Decision-Making | 2 | 2 | 11.5 |

The table above shows that the Education cluster is the theme with the most documents and, according to the strategic diagram, a driving theme. The theme with the highest average number of citations is Ecology, which appears to be a basic and transversal theme. In addition, the theme with the highest centrality, i.e., the most relevant of the clusters, is that of microfinance institutions, followed by education. Regarding the themes with the highest density, i.e., those that are the most developed in their field, we find microfinance institutions and organizational performance.

## 4. Discussion

Despite the remarkable importance of microfinance as an innovative and sustainable tool for poverty alleviation in countries with emerging economies that count on a variety of entrepreneurship for their economic growth, few articles address and examine the relationship between microfinance and entrepreneurship. The present bibliometric analysis was conducted with the aim of understanding how research in these areas has evolved and identifying possible gaps or new topics in the literature, as well as some interesting trends. First, it was shown that the scientific production on this topic had increased significantly in recent years, with the year 2022 being the year with the highest scientific production, suggesting a greater interest of the scientific community in addressing the topic.

Moreover, the results of this analysis agree with those proposed by Liu et al. (2023), who carried out a biometric analysis and concluded that one of the topics in vogue is indeed microfinance as a financial support platform that favors and plays a fundamental role in entrepreneurship. Similarly, the obtained results consider that microfinance institutions, as well as crowdfunding and entrepreneurship, are topics of greater interest in recent years' findings that agree with the bibliometric analysis developed by Coelho et al. (2022), who also suggest that public policy should take into account new trends, such as collective financing through the crowdfunding platform, as a method that allows socioeconomic transformation, poverty alleviation, and empowerment of the vulnerable population. On the other hand, some business topics, such as organizational development and business microcredit, are in line with the research of Rohman et al. (2021), who agree that the literature on microfinance addresses and accompanies these issues, and also includes a topic of financial sustainability, highlighting that microfinance also contributes to this objective.

On the other hand, the work developed by Ali et al. (2022) agrees on some emerging issues, such as education and crowdfunding. However, these authors deepen their analysis in the incorporation of technologies for microfinance institutions with the exclusive purpose of contributing to the eradication of poverty as the ultimate goal of microfinance. In the present analysis, it has been possible to show that microfinance has other business purposes, such as the development of organizations, an economic and ecological approach, and

support for young people and women, among others. Regarding the work of Lwesya and Mwakalobo (2023), one of their topics is gender and equity crowdfunding. However, these authors highlight other issues related to the financial operations of microfinance and the impact of technologies.

In this sense, the results of this analysis allowed locating female microentrepreneurship as a driving theme, which is corroborated by the article proposed by Swapna (2017), who points out that more and more microfinance institutions prefer women as members of access to microfinance because they perceive them as more responsible and reliable; it also shows that investing in women is an effective means in different areas and for society as a whole, and consequently, the need for special support for women in the field of microfinance services is latent. On the other hand, Santos and Neumeyer (2021) also highlight the contribution of microfinance to women's entrepreneurship, as they state that women are the poorest people in the world, which develops in them an entrepreneurial intention that is strengthened by microfinance, thus contributing transversely to the reduction in poverty levels.

On the other hand, another important issue that stands out is crowdfunding, which coincides with the work developed by Chaudhary et al. (2022), who mention that one of the main applications of microfinance and technology are crowdfunding platforms, which allow supporting entrepreneurial initiatives in their initial stage, i.e., entrepreneurship.

It is important to consider that there are few articles that perform bibliometric analysis on microfinance and entrepreneurship, which is confirmed by the results obtained by Nogueira et al. (2020), who systematically conducted their research on microfinance in the context of entrepreneurial finance, using bibliometric analysis to identify the main dimensions of microfinance in academic research. They found that most research focuses on developing countries to promote social and economic development, with a focus on social considerations, economic impacts, and the performance of microfinance institutions. However, financial inclusion and entrepreneurship remain under-explored empirically.

The future lines of research toward which the search and interest of researchers are tending are the main contribution of this article and are Microfinance Institutions, Education, Organizational Development, Entrepreneurship, Microenterprise Credit, and Women's Microenterprise, as they are the most relevant in the strategic diagram obtained (Figure 2). However, Crowdfunding and Ecology are also very close to the category of Drivers, so they should be considered as preliminary research topics. In terms of emerging or disappearing topics, Youth can be highlighted as a topic that can be powerful because it can be developed, considering that the greatest entrepreneurial intention corresponds mainly to young people (Kantis 2016).

Finally, the bibliometric analysis has provided a clearer picture of the literature on microfinance and entrepreneurship and identified some areas where more research is needed. It is hoped that this information will be useful to those working in the field of microfinance and entrepreneurship, as well as to policy-makers seeking to promote economic development and poverty reduction in their countries.

## 5. Conclusions

This bibliometric analysis examined 486 articles from the WoS database and showed that 2022 was the year with the highest scientific production on microfinance and entrepreneurship, with 104 articles addressing the topic from different perspectives, demonstrating the growing interest of the scientific community in addressing the topic. Arvind Ashta is the author with the highest number of publications, with 14 articles, and the journal in which he published the most articles was *Strategic Change—Briefings in Entrepreneurial Finance,* with 35 articles representing 7% of the total number of journals publishing on the topic. The six driving themes that emerged from this analysis were the following ones: Education; Microfinance Institutions; Entrepreneurial Microcredit; Entrepreneurship; Organizational Performance, and Women Microentrepreneurs.

The proposed analysis identified an interesting composition of the strategic diagram, which suggests a broad grouping of themes in the driver section, i.e., with a high internal development, as well as a high relationship with the other themes. The strategic diagram shows that the themes are practically aligned between emerging or disappearing themes and driving themes, a situation that is interesting because it may imply that emerging themes can leapfrog into driving themes and become future lines of the research.

With regard to the driving themes, the literature analyzes them; however, in most cases, in isolation, probably the development of works that link the analysis between microfinance institutions and their products as a support to female entrepreneurship to strengthen the organizational development of these, also promoting education, can encompass these highly developed and correlated themes that can help continue with the objective of microfinance, the alleviation of poverty. On the other hand, the inclusion of topics, such as youth, as support in decision making, and optimal capital structures of this type of institutions or enterprises can also constitute another alternative of analysis that can further strengthen and develop microfinance.

The developed bibliometric analysis is an important contribution to the field of finance and entrepreneurship by examining the existing literature and analyzing the prevailing trends and approaches. In addition, the results of this analysis will enable managers and policy-makers to have a clearer picture of the practices and strategies that have proven effective in the field of microfinance and entrepreneurship development. The research is, therefore, a valuable contribution to providing a solid basis for policy development and implementation of concrete actions to promote access to inclusive financial services and stimulate the growth of entrepreneurship. Thanks to the bibliometric analysis, new opportunities and challenges in the field of microfinance have been identified, laying the groundwork for future research and progress in this area.

Future lines of research can be detailed as follows, with respect to the driving themes: (1) the development and importance of microfinance institutions; (2) the contribution of microfinance to the development of enterprises; (3) business microcredits as an answer to the financing problems of enterprises and SMEs; (4) improvement in the organizational development of enterprises through the use of microfinance resources; (5) support for female microenterprises; and (6) contribution to education, mainly financial education. On the other hand, it is also necessary to keep in mind the following lines of research that can become emerging, according to the strategic map proposed and which are based on the driving themes: (7) contribution of microfinance to the economy; (8) impact of microfinance on green development; (9) contribution of microfinance to youth initiatives; (10) impact of microfinance on capital structures of ventures; (11) contribution of microfinance to SMEs; and (12) impact of microfinance on decision making. Finally, microfinance and technology, through crowdfunding, should not be forgotten as a line of research.

Finally, it can be concluded that the contribution of microfinance is indeed broad, mainly in poor areas or with vulnerable groups; however, it is suggested that this tool can generate business activity in a way that guarantees its sustainability, both as a microfinance institution and as a financial system for enterprises of vulnerable groups. It is suggested to strengthen the line of research, to analyze the impact of microfinance exclusively in female entrepreneurship, and to analyze this influence differentiating developed countries and zones, with some developing ones, to find other factors of influence.

The main impact of this work is the exclusive analysis of the relationship between microfinance and entrepreneurship and the development of future lines of research. This work may have practical implications mainly for microfinance institutions, entrepreneurs, and public policy actors, who should analyze the strong impact that microfinance has on entrepreneurship and promote and strengthen the development of this type of institutions, as well as the strengthening of their products and access to them.

**Author Contributions:** Conceptualization, K.C.-P., J.A.Q., D.H.-T. and P.M.; methodology, J.J.Y. and D.H.-T.; software, J.J.Y. and D.H.-T.; validation, J.A.Q., K.C.-P. and P.M.; formal analysis, K.C.-P.; investigation, J.J.Y. and D.H.-T.; resources, K.C.-P.; writing—review and editing, K.C.-P. and J.A.Q.;

supervision, K.C.-P. and P.M. All authors have read and agreed to the published version of the manuscript.

**Funding:** This article is the result of the ELANET project funded by the European Union and co-financed by the Office of the Vice President for Research at the University of Cuenca.

**Institutional Review Board Statement:** Not applicable.

**Informed Consent Statement:** Not applicable.

**Data Availability Statement:** The data of this study can be obtained from the corresponding author.

**Acknowledgments:** This article is the result of the ELANET project funded by the European Union and co-financed by the Office of the Vice President for Research at the University of Cuenca.

**Conflicts of Interest:** The authors declare no conflict of interest, and the funders had no role in the design of this study, in the collection, analyses, or interpretation of data, in the writing of the manuscript, or in the decision to publish the results.

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
