# Peer review of "Microfinance, an Alternative for Financing Entrepreneurship: Implications and Trends-Bibliometric Analysis"

_ijfs, doi:10.3390/ijfs11030083_

Round 1

Reviewer 1 Report

Congratulations. You have succeeded in presenting a good and interesting article, but there are some things we need to ask.

11. In the abstract it is stated that six driving themes have been identified which should be considered in future analysis. It is better if the themes in question are mentioned briefly without exceeding the appropriate number of words in making an abstract.

22. To sharpen the review of the writing on page 2 paragraph 4 (lines 78-80), it is better to quote one of the articles which describes optimizing financial performance with innovation in financial technology, sustainability and others in the following article. The article in question is; https://doi.org/10.3390/joitmc8010018

33. Why are there two Tables 1 (see rows 111 and 363)

Author Response

Please, find the answers to your comments:

  1. In the abstract it is stated that six driving themes have been identified which should be considered in future analysis. It is better if the themes in question are mentioned briefly without exceeding the appropriate number of words in making an abstract. We have accepted the corrections, mentioning the six driving themes resulting from the analysis.
  2. To sharpen the review of the writing on page 2 paragraph 4 (lines 78-80), it is better to quote one of the articles which describes optimizing financial performance with innovation in financial technology, sustainability and others in the following article. The article in question is; https://doi.org/10.3390/joitmc8010018. We have accepted the review, we consider that the article proposed by the reviewer is suitable for citation.
  3. Why are there two Tables 1 (see rows 111 and 363). We have corrected the numbering of the tables.

Reviewer 2 Report

Please refer the attachment.

Author Response

Please, find the answers to your comments: 

  1. The authors use “Microfinance, an alternative financing for entrepreneurship?” Because the title using question mark, that means the authors have a doubt regarding the function of microfinance, and further want to examine the reality. However, from the article content, it is hard to persuade me that they have already accomplished the purpose. We accept the suggestion by changing the title to the following: Microfinance as a financing alternative for entrepreneurship: Bibliometric analysis of its implications and trends.
  2. In this article, we found two Table 1 and please correct it. Table1: Search indicators. Table 1 summarizes the search indicators used in this analysis. I do believe the researchers or scholars fully understand how to search the related articles they need. Here, put this Table. It seems unnecessary. We have corrected the numbering of the tables, but we feel it is necessary to detail the search criteria of the documents on which this analysis is based in order to justify their use.
  3. The authors mentioned that this similarity between keywords is evaluated using the equivalence index using equation 1. As I know, some software e.g. Turnitin already can do the same thing. While it is true that some anti-plagiarism software such as Turnitin can evaluate the similarity of words, what we do is a co-occurrence analysis of keywords from different articles, not to see similarity between them, but to evaluate relationships.
  4. As for Figure 3. Strategic diagram, I cannot read the words of each quadrant. They are English? Or other country’s language. Figure 3 has been corrected and is in English.
  5. Why the authors choose to use the survey method of Literature review? The author should add some literature review to compare the results of previous studies conducted in the same research context. What is the contribution of the paper to the theory development or other crucial implications? What makes the current study different or standing out from the existing literature? Could the authors give us more explanations to justify your discussions, arguments or concepts regarding the part of methodology. The introduction and conclusions have been expanded, including the reviewer's suggestions.
  6. Normally, we do not put “Results” before the “Materials and Methods”. The authors need to write your article more carefully and logically. We followed the suggestion, although the journal's Word template places Materials and Methods at the end.
  7. After reading the first version of your manuscript, I still cannot fully acknowledge that the article goes into detail enough to provide added value to the Journal of Financial Studies readership. The authors need to clarify the crucial contribution of the research in the Introduction or Conclusion part. It is necessary to clearly state the new concepts and motivating points of the article. The contribution of the work has been clearly specified.

Reviewer 3 Report

The research is a systematic review of the literature in the field of microfinance. A bibliometric analysis is applied, both of the performance of the publications,  as well as a word co-occurrence analysis.

I would like to draw your attention to several elements of the paper, both substantive and technical, that you should correct.

 Although the goal of the research is clearly defined in the Introduction part, you did not briefly state the significance of the research, theoretical, policy makers and managerial decisions contribution. Of course, contributions should be explained in detail at the end of the paper, in the Conclusion. At the end of the introductory part of the paper, it is customary to briefly state the structure of the paper.

 On page 3, line 118, correct the quotation from Rawhouser et. al (2019). The correct citation is Rawhouser et al. (2019). On the same page, lines 121-122 you should also add a period after et al. On page 5, row 160, instead of Lam et al. (2019) asserts, correct to Lam et al. (2019) assert. At the same page, row 162, correct the quote Shkodra et. al (2021) in Shkodra et al. (2021). On page 6, row 186, correct the quote Parvin et. al (2012) in Parvin et al. (2012) and rows 201-202 (Shahzad et. al, 2012) correct the citation. On page 7, row 229, correct the quote (Bruton et. al, 2015). On the same page, rows 229 -300, you wrote the following: "Crowdfunding is based on connecting entrepreneurs and amateur funders through websites". Why amateur funders? People who invest through the crowdfunding platform are backers. On the same page, row 235, correct the source Barguellil1 and Bettayeb (2020) to Barguellil and Bettayeb (2020).

Section Discussions should follow Materials and Methods, not vice versa! On page 10, you listed Table 1! Table 1 was introduced on the third page, on page 10 Table 6 should be inserted. On the same page, row 368, correct the source Cobo et al (2011) to Cobo et al. (2011). Figure 3 should be translated into English.

And finally, it is necessary to expand Conclusions with the following topics: Theoretical contributions, Policy and Managerial Implications, Limitations and suggestions for future research.

Minor editing of English language required.

Author Response

Please, find the answers to your comments:

  1. Although the goal of the research is clearly defined in the Introduction part, you did not briefly state the significance of the research, theoretical, policy makers and managerial decisions contribution. Of course, contributions should be explained in detail at the end of the paper, in the Conclusion. At the end of the introductory part of the paper, it is customary to briefly state the structure of the paper. The introduction and conclusions have been expanded, including the reviewer 's suggestions.
  2. On page 3, line 118, correct the quotation from Rawhouser et. al (2019). The correct citation is Rawhouser et al. (2019). On the same page, lines 121-122 you should also add a period after et al. On page 5, row 160, instead of Lam et al. (2019) asserts, correct to Lam et al. (2019) assert. At the same page, row 162, correct the quote Shkodra et. al (2021) in Shkodra et al. (2021). On page 6, row 186, correct the quote Parvin et. al (2012) in Parvin et al. (2012) and rows 201-202 (Shahzad et. al, 2012) correct the citation. On page 7, row 229, correct the quote (Bruton et. al, 2015). On the same page, rows 229 -300, you wrote the following: "Crowdfunding is based on connecting entrepreneurs and amateur funders through websites". Why amateur funders? People who invest through the crowdfunding platform are backers. On the same page, row 235, correct the source Barguellil1 and Bettayeb (2020) to Barguellil and Bettayeb (2020). The indicated corrections have been made.
  3. Section Discussions should follow Materials and Methods, not vice versa! On page 10, you listed Table 1! Table 1 was introduced on the third page, on page 10 Table 6 should be inserted. On the same page, row 368, correct the source Cobo et al (2011) to Cobo et al. (2011). Figure 3 should be translated into English. The indicated corrections have been made. We followed the suggestion, although the journal's Word template places Materials and Methods at the end.
  4. And finally, it is necessary to expand Conclusions with the following topics: Theoretical contributions, Policy and Managerial Implications, Limitations and suggestions for future research. The conclusions have been expanded, including the reviewer 's suggestions.

Reviewer 4 Report

The paper is interesting. Th writing is clear and concise. The ideas flow logically one after the other. 

I have several recommendations:

- I think the sections should be rearranged. The section with materials and methods should be before the results and discussion.

- the list of references is limited. With a simple search, I found a large number of studies that address the issue of microfinancing for entrepreneurship. I believe the list of references should be expanded and completed with a series of relevant studies.

Author Response

Please, find the answers to your comments:

  1. I think the sections should be rearranged. The section with materials and methods should be before the results and discussion. We followed the suggestion, although the journal's Word template places Materials and Methods at the end.
  2. The list of references is limited. With a simple search, I found a large number of studies that address the issue of microfinancing for entrepreneurship. I believe the list of references should be expanded and completed with a series of relevant studies. Some new studies have been included in the analysis of this paper.

Reviewer 5 Report

This topic is interesting and the outcome of the paper will be a significant contribution to the body of knowledge after some changes/revisions and proofreading. Below are some comments to address. 

The title needs to refined, in its present form it is not fluent. There should be one comma/or full stop/or question mark. 

In the abstract please mention about the time period, from starting year and ending year. 

The introduction section is not well written. It should be started from a broader area/context then narrow down it and relate to your topic/area of study. Highlight gaps and problem there and then proposed solution. Also provide the paper structure therein. The authors should consult: https://www.sciencedirect.com/science/article/abs/pii/S0959652621023179. 

The methodology and methods should be presented before results. Please presents methods after the introduction and then discuss the results after this. 

In methodology, please highlight the time period for which the analysis were performed. Also make a figure to summarize the methods followed, total papers obtained, and relevant papers shortlisted. The authors should consult: https://link.springer.com/article/10.1007/s11356-022-24842-4.

The conclusion should be precise and focused to summarize the study.

Some latest references should be part of this paper.

I hope the comments and papers will help in this regard. 

Minor spell check and proofreading required.

Author Response

Please, find the answers to your comments:

  1. The title needs to refined, in its present form it is not fluent. There should be one comma/or full stop/or question mark. We accept the suggestion by changing the title to the following: Microfinance as a financing alternative for entrepreneurship: Bibliometric analysis of its implications and trends.
  2. In the abstract please mention about the time period, from starting year and ending year. The indicated corrections have been made.
  3. The introduction section is not well written. It should be started from a broader area/context then narrow down it and relate to your topic/area of study. Highlight gaps and problem there and then proposed solution. Also provide the paper structure therein. The authors should consult: https://www.sciencedirect.com/science/article/abs/pii/S0959652621023179. The introduction and conclusions have been expanded, including the reviewer 's suggestions.
  4. The methodology and methods should be presented before results. Please presents methods after the introduction and then discuss the results after this. We followed the suggestion, although the journal's Word template places Materials and Methods at the end.
  5. In methodology, please highlight the time period for which the analysis were performed. Also make a figure to summarize the methods followed, total papers obtained, and relevant papers shortlisted. The authors should consult: https://link.springer.com/article/10.1007/s11356-022-24842-4. We have accepted the corrections
  6. The conclusion should be precise and focused to summarize the study. The conclusions have been expanded, including the reviewer's suggestions.
  7. Some latest references should be part of this paper. Some new studies have been included in the analysis of this paper.

Round 2

Reviewer 1 Report

Thank you very much for the response and improvement made to the article.

Reviewer 2 Report

I consider this article can be going to publication, because the athors already revised the paper according to the reviewers' comments.

I consider this article can be going to publication, because the athors already revised the paper according to the reviewers' comments.

Reviewer 3 Report

Your paper looks significantly better than the first version. You have accepted all my suggestions and I think that your paper has enough quality to be published.

Reviewer 4 Report

The authors took into account my recommendations.

They added only 3 studies to the reference part.

Reviewer 5 Report

The revised version looks fine. The authors have tried to address the comments.